# Local-Indigenous Autonomy and Community Streetscape Enhancement: Learnings from Māori and Te Ara Mua—Future Streets Project

**DOI:** 10.3390/ijerph18030865

**Published:** 2021-01-20

**Authors:** Kimiora Raerino, Alex Macmillan, Adrian Field, Rau Hoskins

**Affiliations:** 1Ngāti Awa & Ngāti Rangiwewehi, SHORE & Whariki Research Centre, Mackie Research & Massey University, Auckland 1010, New Zealand; 2Department of Preventive and Social Medicine, University of Otago, Dunedin 9054, New Zealand; alex.macmillan@otago.ac.nz; 3Dovetail Consulting Ltd., Auckland 1245, New Zealand; adrian@dovetailnz.com; 4Ngāpuhi, DesignTRIBE Architects, Auckland 1021, New Zealand; rau@designtribe.co.nz

**Keywords:** Indigenous, co-design, streetscapes, re-indigenisation, indigenous autonomy, Māori

## Abstract

In settler countries, attention is now extending to the wellbeing benefits of recognising and promoting the Indigenous cultural identity of neighbourhoods as a contributing factor to more equitable and healthier communities. Re-indigenisation efforts to (re)implement cultural factors into urban design can be challenging and ineffective without the leadership and collaboration of local-Indigenous peoples. Undertaken in Aotearoa New Zealand, Te Ara Mua — Future Street project, demonstrated that co-design has critical potential in the reclamation of Indigenous autonomy, increased local-Indigenous presence and revitalisation of cultural identity. Employing a Kaupapa Māori (Māori-centred) research approach, we focused on the workings and perspectives of mana whenua (local-Indigenous peoples) and community stakeholder engagement in Te Ara Mua. An Indigenous theoretical framework, Te Pae Mahutonga, was utilised in the data analysis to explore perspectives of Indigenous collective agency, empowerment, and wellbeing. Our research demonstrates that developing capacity amongst Indigenous communities is integral for effective engagement and that the realisation of autonomy in urban design projects has broader implications for Indigenous sovereignty, spatial justice and health equity. Significantly, we argue that future community enhancement strategies must include not only re-designing and re-imagining initiatives, but also re-indigenising.

## 1. Introduction

Nōku te whenua, o ōku tūpuna—The land is mine, inherited from my ancestors Māori *whakatauki* (proverb).

Community streetscape enhancement demonstrates the intention of cities to redesign and enhance existing public spaces that influence peoples’ health and wellbeing. Central elements of community retrofitting include increasing the everyday movement of people between their living, work and leisure environments, and making communities healthy. Interlinked health, social and environmental challenges such as obesity, diabetes, climate change, and rising social and health inequities continue to affect individual, family and community wellbeing [1,2]. Compounding these issues are the health effects of increased car reliance and a reduction in active transport among individuals and families [3,4]. While many strategies are required to address these ongoing challenges, the design or redesign of streetscapes offers significant potential to improve wellbeing within people’s homes and communities. As an important example, community wellbeing can be enhanced by making public spaces (e.g., streets, walkways and parks) safer and more attractive, thereby encouraging people to walk, cycle, take public transit and socialise [5,6]. 

In settler countries such as Aotearoa New Zealand and Canada, attention is now extending to the holistic benefits of promoting the Indigenous cultural identity of urban communities [7,8]. This includes cultural landscaping or placemaking initiatives that reaffirm Indigenous identity and visibility, contributing to a sense of connectedness and belonging for all community members. However, past and recent studies have shown that efforts to (re)implement cultural identity into community redesign can be challenging and ineffective without the leadership and collaboration of Indigenous peoples [9,10,11]. In this article, we argue that the exercise of local-Indigenous autonomy in New Zealand is a fundamental concept and embedded treaty right in the re-designing of neighbourhood physical environments for health, equity and sustainability.

To understand constructions of Indigenous autonomy in urban redesign, it is important to first clarify our application of the terms ‘re-indigenisation’ and ‘local-Indigenous autonomy’. First, situated amid ongoing discourse regarding conceptualisations of indigenisation and decolonisation [12,13,14], we employ ‘re-indigenisation’ to signify the workings of Indigenous peoples to reaffirm their enduring presence and culture on their traditional lands. Re-indigenisation restores, recognises and empowers the first occupants of a community in situ [15,16]. As Indigenous design approaches develop, we define re-indigenisation as an Indigenous-led movement to reclaim everyday living environs by contesting, unsettling and disrupting the ongoing creation of urban centres that serve only to reflect settler power [17,18,19]. Historically, a legacy of British property legislation [20] has entailed defined boundaries, renaming places, and individual privatised land ownership which formed the basis of emplacing settlers and eradicating Indigenous societies [18,21]. Matunga [22] and Nejad [11] concur that this legacy of colonisation and urban development on Indigenous lands has entailed the elimination of Indigenous ‘memory’ (existence, heritage, experience) and ‘materiality’ (physical presence, structures, places). The dispossession of Indigenous peoples has compelled long-term reclamation and re-indigenisation strategies that continue to counteract ongoing colonial processes still prevalent in urban design.

Our second term, ‘local-Indigenous autonomy’, denotes the concerted efforts and practices of area specific Indigenous groups to protect and care for their peoples and traditional lands, by reinforcing their ownership, control and rights [23,24,25]. Rather than the overarching or broad approach of ‘Indigenous autonomy’, this localised practice is tribally-led and driven by historical knowledge and intimate experiences of their land, people and culture. Yet, worldwide the exercising of local-Indigenous autonomy has faced a myriad of barriers and challenges over many decades [26,27,28]. These issues comprise of a mix of colonisation and urbanisation forces, and inter-tribal rural and urban disputes [29]. Academics claim that cities can be termed ‘highly contested sites’ as urban issues between Indigenous and non-Indigenous peoples are more complex, prevalent and overt [30,31,32]. For example, despite a significant number of now urban Indigenous residents globally [33,34,35], there remains little acknowledgement or implementation of local-Indigenous autonomy. Settler cities today, reflect sites of oppression that perpetuate loss of agency while weakening land, cultural and identity connections [11,36]. The loss of Indigenous agency has resulted in increased calls for urban-Indigenous peoples to reaffirm their rights, identity and co-presence, and to reclaim their right to their city [37,38,39]. International Indigenous rights affirm that Indigenous peoples are not mere participants or stakeholders in their communities [40,41], but in actuality, leaders and partners that have an important role in informing planning, design and community development [11,36,42]. 

In New Zealand, the exercise and expression of local-Indigenous autonomy is gaining momentum as Māori engagement has increased in design projects [7,43,44]. For example, the ongoing rebuild of the earthquake damaged city, Christchurch, has entailed a ‘cultural recovery’ that is reliant on a working partnership with Ngāi Tuahuriri the local *hapū* (All Māori terms are italicised and explained on the first instance only) or subtribe of the traditional Māori land occupants known as mana whenua [36]. Mana whenua refers to local hapū who retain *mana* (traditional authority) over their *whenua* (land) and traditions [43,45]. Succinctly, mana whenua are local-Indigenous traditional land owners. The role and responsibilities of mana whenua centre on *iwi* (tribe) and hapū affairs, including advocating for tribal wellbeing and fostering community relationships. Mana whenua have rights and responsibilities that strengthen their autonomy, based on their ancestral land occupation and recognition in New Zealand’s founding document, te Tiriti o Waitangi (The Treaty of Waitangi) [46]. The signing of the Treaty in 1840 between Māori and the Crown established the right of settler governance while guaranteeing Māori ongoing sovereignty and retention of cultural, social and physical resources [47]. While ongoing tensions inherent in the wording and exercise of te Tiriti are not explored here [48,49,50], for mana whenua, the treaty conferred rights of access, execution, control, and authority in decision-making processes, as well as equity of outcomes for health and wellbeing [51]. Notwithstanding, mana whenua leadership and rights has endured prolonged contestation that only in recent years has seen their gradual recognition and inclusion into community projects. This increased engagement between mana whenua, local authorities and urban planners has resulted in the exponential inclusion of Māori in decision making and the recognition of Māori values and knowledge in design planning and policy [52]. Further, as mana whenua engagement has increased, Māori cultural landscaping opportunities have emerged, contributing to the conception and application of Te Aranga Design Principles, a set of outcome-based principles founded on intrinsic Māori cultural values [53,54] (see further Section 2.2) that can be applied to urban design strategies. 

Both re-indigenisation and local-Indigenous autonomy are empowered practices that counteract historical and prevailing settler colonising processes that enabled neighbourhoods to be voided of their histories and landmarks [22,31], and thereby their very presence. Crucial to these Indigenous-led practices is the recognition and re-enactment of Indigenous rights and obligations [11,55], not only to participate in tribal and community affairs, but also in wider local governance and decision making for community planning. For Māori, the greater potential of redesigning urban streets is more than Indigenous cultural aesthetic promotion. At the centre here, is the fulfilment of governmental obligations and the rights of Māori as per te Tiriti o Waitangi and more broadly the United Nations Declaration on the Rights of Indigenous people. The fulfilment of *tino rangatiratanga* (autonomy) and *ōritenga* (equity) is specifically guaranteed in Article II and III of te Tiriti. Equitable local-Indigenous engagement, processes and outcomes is fundamental to improved urban-Indigenous communities and reducing health and social inequities. 

In this article, we explore these concepts of re-indigenisation and local-Indigenous autonomy in the community retrofit project of Te Ara Mua — Future Streets (Te Ara Mua) in Tāmaki Makaurau (Auckland), Aotearoa New Zealand. This project provides a recent case study of mana whenua and community stakeholder perspectives regarding their relationship in collaborative urban-Indigenous design. The aim of Te Ara Mua was to retrofit street changes at a suburb scale that fostered multiple health, cultural, social and environmental outcomes.

## 2. Methodology 

The methodology underpinning this study is Kaupapa Māori. This Indigenous research approach was used because it privileges Indigenous voices and lives, while drawing on cultural narratives and concepts to provide culturally relevant understandings of specific phenomena. Fundamental to this approach is that the research is led by Māori for Māori, contributing to Māori wellbeing gains and advancing positive Māori development [56,57]. Employing a Kaupapa Māori research approach entails a decolonising agenda that ensures Māori ideologies, practices, and aspirations determine all phases of the research [58,59,60]. Accordingly, the analytical framework for this work is guided by Dr Mason Durie’s [61] Te Pae Māhutonga model for achieving optimal Māori health. Wellbeing within this model requires access to a secure cultural identity, access to land, language and cultural resources and the ability to participate fully in society, alongside healthy lifestyles and a healthy (living) environment [62,63]. Te Pae Māhutonga can be applied across multiple disciplines centred on Indigenous wellbeing including social, cultural and environmental arenas. Important to this research, Te Pae Māhutonga analysis is culturally responsive underpinning the promotion of autonomy and participation for Māori individuals and whānau (family). Utilisation of this Indigenous framework enables a broader view of wellbeing that is holistic and reflects Indigenous realities [64,65]. We used qualitative case study methods [66] to undertake a process evaluation of Te Ara Mua project. Ethical principles for conducting Kaupapa Māori research have also been proposed (for example see L. Smith) [60]. In the case of this research, these included respect for people; sharing, hosting and generosity; being cautious; upholding and respecting mana whenua knowledge; and prioritising mana whenua voices in reporting.

Two authors (KR and RH) are experienced Kaupapa Māori researchers. The data collection and analysis were undertaken by the first author (KR), who was not part of the co-design process, or the research team during the co-design phase. The other authors (AM and AF) are *Pākehā* (New Zealand settler of European descent) and *tangata Tiriti* (people in NZ by virtue of te Tiriti o Waitangi, with a te Tiriti-mandated responsibility for addressing colonial institutions and policy). AF and RH were both part of the co-design process, and therefore can be seen as “insiders” in this research, while AM is the research lead for the Te Ara Mua project. These varying roles in the co-design process and this research mean that our analysis covers a spectrum of gazes and reflexive positions. 

### 2.1. Case Study Setting

Te Ara Mua is a controlled before–after intervention study of neighbourhood street changes to make walking and cycling safe and easier, and reflect the cultural identity of Ngā hau Māngere known today as Māngere, in Auckland, New Zealand. At the core of the project was the hypothesis that, in suburban communities, a wide range of social and health benefits can accrue by increasing active transportation, (e.g., walking or cycling), by retrofitting environments to safely promote active travel, and conversely de-prioritising private motor vehicle travel. The evidence supporting this study, the collaboration design of the intervention, and the research study design have all been previously published [67,68]. Both placemaking and transport infrastructure strategies were employed. Project outcomes included ensuring that streets and lanes were accessible and user friendly to walk through, and the best thoroughfares were also attractive destinations in their own right. The project setting, Māngere, comprises of 70,959 [69] residents of which 60% identify to a Pacific Islands ethnic group, followed by 20% European, 17% Asian and 16% Māori (includes all people who stated each ethnic group, whether as their only ethnic group or as one of several) [70]. While Māngere is a predominantly Pacific Island community, the first peoples are Māori represented by the mana whenua of Te Ākitai Waiohua and Te Ahiwaru Waiohua [71]. Other mana whenua groups for Māngere and the wider Tāmaki Makaurau region include: Ngāti Whātua Ōrākei, Ngāti Te Ata, Ngāti Pāoa, Ngāi Tai and Kawerau-a-Maki [72]. It is important to note that while tribal identification of mana whenua within Tāmaki Makaurau is relatively straightforward, identifying set tribal land boundaries is more complex. Similar to other tribal groups globally, there remains unresolved mana whenua histories of inter-tribal land disputes. In this regard, project settings and inclusion of relevant residing mana whenua groups is often more inclusive rather than exclusive. Figure 1 identifies the project boundaries of the site.

### 2.2. Mana Whenua Engagement

Engagement for this project involved nine meetings between mana whenua representatives (5) and wider community stakeholders, including urban designers (3), Auckland Council (2), New Zealand Transport (2), and Te Ara Mua project members (6). These meetings were chaired by project lead, Rau Hoskins (co-author), a leading expert in the field of Māori architecture and cultural landscape design. A total of five mana whenua members attended various project meetings, and three of this group attended between 3 and 7 meetings. Of this number two were respondents for this study. Several pre-engagement meetings were conducted on marae and community venues, to engage interest and participation from local mana whenua and Māori community groups. Mana whenua engagement centred on the application of Te Aranga Design Principles to streetscape enhancements to part of Māngere Central. Stakeholders and mana whenua representatives discussed and reviewed cultural landscaping outcomes that honoured and represented mana whenua narratives, presence and sites of cultural significance. Led by mana whenua through providing advice and local knowledge, cultural initiatives were designed and delivered that enabled deeper sociocultural place connections for Māori and non-Māori community members. Broadly, this cultural design strategy includes both tangible and intangible processes and outcomes that aim to reinvigorate Māori cultural identity and autonomy with their communities while recognising treaty obligations and addressing inequities. Below, Figure 2 and Figure 3, demonstrates the workings and implementation of Te Aranga Design Principles into Te Ara Mua project. Figure 3 displays images (photos 1–6) of the five design outputs implemented into the streetscapes of Māngere Central.

### 2.3. Study Participants

The respondent group for this study consisted of four panel members and one Māori artist (see below Table 1—interview participant ‘X*’). Although the Māori artist had not participated in the design panel, he played a central role in the implementation of the design through commissioned work, and provided unique insights as a Māori public artist (see Photo 1), local resident and mana whenua member. Reflective of the current capacity of mana whenua and the requirements of street design projects (as discussed in Section 4), the ten panel members consisted of only four Māori members and six non-Māori. All the Māori members had worked on other design projects and were familiar to each other. Underpinning the collaborative relationship between the urban designers and mana whenua was the implicit necessity of developing the current knowledge and capacity of mana whenua. In this sense, the panel meetings created a relational space for enriching Indigenous collaborative processes, discussions and outcomes. 

All of the participants gave written permission to be identified in the utilisation of information gathered from interviews. Ethical approval was obtained from The University of Auckland Human Participants Ethics Committee, The University of Auckland Research Office, New Zealand (18/010723).

### 2.4. Data Collection and Analysis

All non-research team members of the co-design panel were invited to take part in individual (face-to-face) in-depth semi-structured interviews were conducted between January and June 2019, approximately 2.5 years after the completion of the co-design process. Each respondent selected their interview site including work and café settings. The interview discussions centred on two broad questions: ‘What was your contribution to the mana whenua engagement panel?’, and ‘What do you consider were the challenges and successful aspects of this engagement?’ Follow-up questions varied, dependent on the role and expertise of each respondent, yet remained centred on mana whenua engagement and outcomes. All respondents were offered a *koha* (gift, offering) for their generosity in sharing their time and knowledge for this study. Interviews were digitally recorded, transcribed, and sent to each respondent for review. 

Data analysis was guided by Durie’s Te Pae Māhutonga framework, which served as a base for the manual coding. This framework has been developed specifically for Māori and provides an overarching conceptual tool for analysing specific environmental-based wellbeing situations. Drawing on health promotion indicators that contribute to health and wellbeing, this framework comprises of six ‘enabling’ markers for examining the complex interrelationships underpinning holistic wellbeing: Mauriora (cultural), Waiora (physical environment), Toiora (community health), Te Oranga (society participation), Ngā Manukura (leadership and relationships), and Te Mana Whakahaere (autonomy and control). This Indigenous framework is a flexible analytic tool that enables a hybrid qualitative approach of deductive classification and organisation of data into defined markers, before a more inductive analysis that identifies themes within each marker [73]. Utilising thematic analysis [74], we explored the reflections and experiences of the respondents of mana whenua engagement in the Te Ara Mua project. This was done through a mix of inductive and deductive analytic approaches [75] which enabled the identification of key patterns across the respondents’ accounts. Initially, our deductive approach focused on broad Māori wellbeing concepts, then we identified latent inductive constructs or sub-themes. This dual analysis involved our Kaupapa Māori researcher (KR) repeatedly reading the transcripts, and coding the entire dataset into the demarcation of themes. An initial long list of themes was drawn from the coded dataset. At this point, all the themes were reviewed and discussed among the authors, and further refinement was undertaken until a final set of themes was identified and agreed on. The quotes presented have been edited to condense salient points. However, the original meaning of the quotes have not been altered. Maintaining authenticity of respondent meaning is crucial in kaupapa Māori research and amplified by respondent agreement in this study not to be anonymised. 

## 3. Results

Four co-design panel members (two mana whenua and two stakeholders), as well as one of the mana whenua artists involved in the project, were interviewed: a total of five respondents. Consistent with a Kaupapa Māori research approach, priority has been given to Māori understandings and mana whenua voice. As a result, all main quotations are mana whenua perspectives only, as a deliberate strategy to honour and develop Indigenous knowledge. Notwithstanding, all participant interview data (Māori and non-Māori) were reviewed and determined the study themes. Our results focused on perspectives of the processes and outcomes of mana whenua engagement. The mixed deductive and inductive approach led to an initial set of 19 sub-themes, which we amalgamated and refined to a set of themes that were organised into the broader themes of Te Pae Mahutonga. This resulted in five overlapping themes, representing Māori (environmental) wellbeing understandings. Aligning to the framework indicators of Te Pae Māhutonga, we identified these five themes: (i) embedding Te Ao Māori principles in urban design (Mauriora); (ii) Māori (presence) in place (Waiora); (iii) healthier communities for all (Toiora); (iv) Māori (participation) in design projects (Te Oranga); and (v) reasserting mana whenua leadership and autonomy (Ngā Manukura and Te Mana Whakahaere). Respondents’ discussions consistently placed their experience as part of Te Ara Mua in the context of a longer history of experiences with urban design projects. We have included these more general experiences in the discussion of each theme, differentiating where needed.

### 3.1. Embedding Te Ao Māori in Urban Design

All of the respondents commented on varying aspects of Te Ao Māori (Māori worldviews) principles and urban design linked to Māori and community wellbeing. This influence was aptly described by Mei, commenting that “seeing ourselves, our values, our *tohu* (symbols) including *te reo* (language) reflected back is vital to our wellbeing and identity.” Among the respondents was a consensus that applying Māori principles visually in public art was a necessity, but also in behind-the-scenes design processes. Essentially, that these processes must be embedded throughout all stages of design engagement including recruitment, meetings and dissemination. *Karakia* (prayer) was identified as an example of a necessary Māori practice utilised in meetings to ensure a culturally comfortable, respectful and safe environment for all. Berenize succinctly aligned this observance of Te Ao Māori principles with cultural awareness, she argued, “you are not showing me any respect for who I am or the people that I represent, if you don’t know these very small ways—our *tikanga* (protocols) of how things are done.” Reviewing past design projects, Mei added that inherent difficulties arose from engaging with non-Māori stakeholders who lacked understandings of Māori perspectives, practices and values. Berenize and Mei spoke similarly that this was not an issue for Te Ara Mua due to the inclusion of a Māori co-investigator, Rau Hoskins (Ngāpuhi, co-author). The respondents acknowledged previous work undertaken with Rau, and stated his involvement and expertise motivated their engagement and, for mana whenua, provided an implicit assurance of the inclusion of, and respect for, Te Ao Māori principles. 

Together, mana whenua respondents commented on Te Ao Māori practices of *manaakitanga* (sharing), *kaitiakitanga* (caring), *whanaungatanga* (connectivity) and *rangatiratanga* (leadership) as cultural guidelines and influences for their engagement in design projects. These cultural practices reinforced their role and responsibilities as project members, and the intended design outcomes. Respondents commented that Te Ao Māori practices could be further incorporated and highlighted in public artwork. For example, drinking fountains as a practical medium for promoting Māori principles of sharing, caring and amongst community members, as explained by Berenize: “for mana whenua, we are about *manaaki tangata* (caring for people) … we’ve got all of these visitors that come here every day—at the very least we should be able to provide them with clean drinking water.” Notwithstanding, mana whenua respondents acknowledged that challenges remain in the equitable incorporation of Te Ao Māori principles in design projects more generally with more effort required from community stakeholders, despite marked improvement in recent years.

### 3.2. Promoting Māori in Place

The physical environment of the intervention area and the visual design outcomes were reviewed (photographs supplied—see Figure 3) by all respondents. These photographs of the project design outputs were utilised as prompts or reminders of the tangible project outcomes and their placement within the community of Māngere, including native planting and public artworks. The visual outputs were explained and praised by each respondent as they described their personal design and development experiences in the Te Ara Mua project. Respondents commented that Māori public artworks remain an effective and crucial method of promoting Māori presence in place. Reflecting on the project pou, respondents noted they were crucial public acknowledgements and statements of local iwi, and a means of anchoring Māori to place. The notion of anchoring was explained by Mei that “It’s an important part to iwi, hapū, and whānau wellbeing being able to see yourself reflected back in the public realm and anchoring your identity through *whakapapa* (genealogy) in your special place in the world.” Wider community stakeholders, Yoko and David, viewed pou as tributes to celebrate Māori and their historical presence. Local resident, Chris explained that his pou had personal significance for his whānau and iwi, “they were really proud, because they had their ‘own’ doing those carvings. You could see it on their faces … they looked really happy … it was [a] good thing for the families.” Other respondents noted that Māori public artworks also contributed to increased cultural education, ownership and connection to place for all community members. This interrelated concept was detailed by Mei: 

“Our *toi* (art) has a real role to play with educating all. It is that *mātauranga Māori* (Indigenous knowledge) being expressed in a way that is accessible to others. That they understand. They learn more about Māngere. They learn more about mana whenua or they learn more about important features out in that landscape.”

Considering future project work, several respondents reinforced the need to include cultural narratives. For example, wayfinding signage that provided cultural narratives including: identification of mana whenua; significant landmarks and artwork; and local *kōrero* (information) for visitors and locals. Lastly, mana whenua respondents each acknowledged that there remained a lack of visible Māori presence in Tāmaki Makaurau. These respondents hoped future design projects would include significant Māori architectural statements that promoted mana whenua presence.

### 3.3. Healthier Communities for All

The link between healthier communities and urban design was discussed by all of the respondents, specifically the benefits of physically active, cultural and social experiences. Making their community environments more pleasant to walk in and safer to partake in physical activities was a strong discussion point among all respondents. Respondents acknowledged that good design must enhance social and cultural connections for improved wellbeing outcomes for all residents. Design and sociocultural connectivity was described by David, that “for people’s spirit and sense of community … it’s about making space humane and enjoyable across communities … it is revealing something about a place and making you feel at home in it.” Mana whenua respondents agreed that focusing on the health and safety of community members, tamariki in particular, was a responsibility for Māori as kaitiakitanga in practice. Expanding this point, Mei argued that designing healthier communities entails a broad holistic view, involving mana whenua leadership and wider considerations of wellbeing implications, including environmental:

“[as] *kaitiaki* (caretakers) just being able to say within urban development: “Where is the *paru* (dirt/sewage) going, and is it going into our *moana* (water) Have you thought about the contamination? Are you being sustainable with your building materials?” So that is what I also see as active kaitiakitanga - being able to protect the *taiao* (environment). The biggest narrative right now, pulsating around the world is climate change, that is the grand narrative, and mana whenua are truly tuning into that in a better way.”

Again, mana whenua respondents all noted considerations of their role as kaitiaki was a major influence to their participation in design projects. Yet, it is important to note that the exercise of *kaitiakitanga* (caretaking) is restricted by colonial limitations linked to privatised land ownership and autonomy issues. 

### 3.4. Empowered Māori Participation in Community Design

Māori participation in local design and development was widely discussed among the respondents. Specifically, the necessity of mana whenua to advise, participate and lead in decision making concerning community design or redesign. Capitalising on the local and historical knowledge of mana whenua was identified as crucial for cultural design initiatives. As the respondents reflected on their participation in other projects and Te Ara Mua there was agreement that further evolvement of mana whenua engagement is required. Firstly, earlier engagement in projects, starting at design procurement and planning briefs. Two mana whenua respondents recalled prior projects, in which their involvement had commenced only after the design plans had been determined. Correspondingly, Yoko commented that time restraints, limited availability and capacity often resulted in later engagement which needed future review. Together, respondents posited that more in-depth or broader understandings of an overall project would be useful, and for the Te Ara Mua project design workshops would have been useful to increase iwi knowledge and expertise. Several respondents commented that earlier engagement and all projects would contribute to a more empowered approach of mana whenua leadership.

Secondly, both mana whenua and community stakeholders noted that Māori participation needed to be meaningful. Meaningful engagement was signaled as equitable, respectful, trust-worthy and based on genuine relationships, not ‘lip service’ or ‘ticking a box’. Reflecting on past experiences, Berenize argued the distrust occurs when designs outcomes have been predetermined with limited engagement, she explained: 

“Engaging with XXX … show[ing] us three completed designs and then wanted us to choose one … They had ‘A’ ‘B’ and ‘C’ … “but the one we [they] really like is B”. What they really wanted was for us to rubber stamp their decisions and to be able to say that collectively we worked on a design for a building. I was like “nah, Ngāti Te Ata doesn’t not support that–this is not how designing anything works” … we walked away from that, because that isn’t anything apart from the rubber stamp.”

Similarly, hidden agendas and limiting Māori input were identified as past design experiences by mana whenua respondents. These respondents challenged future projects to ensure meaningful engagement developed from better communication, shared learning, and agreement of design outcomes. Utilising mana whenua participation as an opportunity for mutual learning experiences was highly regarded among respondents, such as learning regarding the site and its local cultural layers. Yoko described ‘cultural layers’ as Māori understandings and knowledge of human histories, natural and geological features of their urban settings. Aside from mana whenua and community stakeholder engagement, many of the respondents also acknowledged that the internal relationships amongst differing mana whenua groups need further development. Internal tensions among differing mana whenua groups was noted as a challenge for effective participation, including the tensions of working with other mana whenua or iwi groups and conflicting opinions of mana whenua representation in public spaces. Setting aside mana whenua differences and difficulties was discussed by all the mana whenua respondents as essential. Mei summarised current mana whenua relationships, noting that “trying to get agreed narratives is not exactly easy, but it is more about we have all got a history. Each iwi can probably give their own kōrero about a particular time on that timeline where they were present and lived, worked and played … we do agree to agree, and we do agree to disagree.” The most important factor of Māori project participation is increased voice and visibility, as Mei explains further:

“I think that it is vital that we are at the table. We give a layer of history that you just can’t take away from us, you can’t separate. We are just intrinsically knowledgeable about particular sites, and depending where the project is—it is important that we do contribute, because of our knowledge, our identity, being *ahi kā* (original occupants)–they are not going to be going away.”

The third and last point advocated by the respondents was the necessity of pushing the current scope of design engagement. Extending current design engagement for Te Ara Mua entailed more involvement beyond cultural consultation and input, as noted by Mei: 

“We can put something on the bridge, but we actually want to part of designing the bridge, the form of the bridge—not just sticking on a motif. I think we have progressed from ‘patterns in the pavements’.” 

While acknowledging that design engagement had evolved from consultation to engagement, there was universal agreement that co-design is an essential goal. Similarly, Yoko reflected that small developments have occurred, stating that “the evolvement of the process is already changed from consultation to engagement, but perception has changed already, but how we change that culture of engagement now to co-design.” Together, the respondents agreed that while much progress had occurred regarding mana whenua participation, further progress is necessary.

### 3.5. Reasserting Mana Whenua Leadership and Autonomy

The predominant theme among the respondents was mana whenua leadership and autonomy in urban design projects. Universally shared among the respondents was the correlation of mana whenua leadership, good community design and wellbeing. All mana whenua representatives spoke of the importance of autonomy and control of their lands. Autonomy was linked to Māori and non-Māori wellbeing derived from a sense of ownership and belonging to their communities. Together, mana whenua representatives agreed that their iwi were increasingly assertive and progressive regarding promoting wellbeing opportunities in their communities. Hence, increased mana whenua autonomy is essential, as Mei aptly summarised:

“We don’t want to be in a queue, and you will hear that from mana whenua, ‘we are not in a queue, we are a partner, we are not a stakeholder’. It is inevitable that we have gone through that phase of a hundred and odd years of being excluded and now I believe that mana whenua have a role to reassert our cultural design and narratives—that can then inform the look and feel of our spaces.” 

Discussions detailed engagement experiences, and an agreement that developing mana whenua capacity for design projects was integral. Respondents noted that as community design projects increase, so too have the demands for mana whenua leadership and expertise. Chris identified current capacity issues within his iwi, including small membership and limited availability. Yet, Berenize posited that recent growth of engagement in design projects has, in contrast, resulted in iwi capacity development. She explained her iwi development, “because they learnt that at this [project]. Then they go to the next place and they go— ‘well these guys are doing this, why don’t you try that’—they are improving every project.” Nonetheless, design workshops that developed Māori capacity was highlighted as crucial by urban designer Yoko and Berenize. Summarising the discussions of developing iwi capacity, Mei speculated on current potential to move beyond project engagement to further co-design opportunities such as Te Ara Mua: 

“Mana whenua have to build our capacity, to have the right ones onboard to be able to co-design in the end. One of the challenges right now for mana whenua—is building the capacity of our artists and designers, to bring them onboard and introduce them into a project. So that they can co-design as well, be right there.” 

Each of the respondents acknowledged that they were pleased with both their design engagement experiences and the physical outputs of Te Ara Mua, and looked forward to future co-design opportunities. Mei reflects the view of all the respondents’ in reinforcing the importance of mana whenua leadership in urban design: 

“We have a lot to contribute and I think it is vital that we are across and contributing to urban design and development. We have a different lens that is important to our own sense of identity and indigeneity, as I said if we don’t see ourselves we lose sight of it and we just kind of blend in with all the other nationalities and that should not happen.”

## 4. Discussion

Primarily, our results reaffirm the desire of Indigenous peoples to build long-term Indigenous design capacity [52,76]; reclaim Indigenous-urban spatial narratives [38,42]; reassert local-Indigenous participation and autonomy; and promote local-Indigenous distinctiveness [45] by going beyond tokenistic measures [42,77]. While achieving these outcomes is reliant on protecting and nurturing an equitable working partnership (Article III of te Tiriti), improved engagement and greater Indigenous autonomy has the potential to create positive, inclusive and innovative design outcomes. All of the respondents showed that co-design projects have critical potential to be instruments in the reclamation of Indigenous autonomy, increased local-Indigenous presence and revitalisation of cultural identity. 

Indigenous-urban design potential has been highlighted, not only with the development of cultural landscapes within urban settings, but more importantly the participation of local-Indigenous peoples’ as leaders, influencers and decision-makers in the redesign of their lived communities. The results emphasize the necessity of local-Indigenous autonomy in urban design, as a re-indigenising equitable approach that recognises and empowers first peoples. This approach makes an important contribution to increased Indigenous voice and visibility by reaffirming their presence and ownership of traditional lands despite the processes of urbanisation and colonisation [32,39]. Indigenous public artwork was viewed as contributory to Indigenous reaffirmation and revitalisation strategies, creating a symbolic capacity, and reassertion of cultural connections to land and place [37]. Yet, as noted by the respondents, the application of visual elements is not itself sufficient, it is about local-Indigenous peoples creating sociocultural connections with place [44] by drawing from Indigenous knowledge and histories. Local-Indigenous peoples provide unique local wisdom that can inform design projects contributing to the co-creation of more equitable, sociocultural, safer and walkable public spaces.

This is the first study to explore mana whenua engagement in urban design. The results of this study support and strengthen recent literature regarding Indigenous-urban design in Aotearoa [36,38,76], by contributing new insights and findings supporting re-indigenisation strategies. Of particular significance is the capture of new qualitative data in the relational space, from mana whenua and community stakeholders. The in-depth nature of the interviews and expertise of the respondents ensured rich qualitative data that provided important insights into Te Ara Mua project engagement. In addition, the willingness of our respondents to be identified here, their iwi and organisations named, is significant and adds further weight to the validity and reliability of our results. 

There were some limitations to this research that should be considered in evaluating the findings. First, all the interviews took place only once and that multiple interviews may have led to more in-depth responses. Second, the sample size of five was small due to the limited respondent pool and the over commitment of mana whenua panel members, limiting the generalisability of the results. However, the data were rich, revealing in-depth discussions and reflections of Te Ara Mua and its effectiveness. Because this research is about the relationship between mana whenua and colonial-settler governance institutions, representatives of both were included in our interview sample. Issues of representativeness volumes were considered less important than our objective of obtaining views from people that were active members of the project design group. Last, although the study scope was limited to one project and area in Tāmaki Makaurau, it has promising potential for other urban settings both regionally and internationally. 

This research, while specific to a single co-design process and a small number of mana whenua voices, highlights the fundamental importance of the role of local-Indigenous leadership and knowledge in the design processes. Revealed in this case study were some pivotal drivers linked to re-indigenisation that served to influence design engagement by mana whenua. The reflections of mana whenua respondents were overwhelmingly positive; specifically, respondents’ discussions of the project relationships and outcomes were identified as meaningful and effective design engagement. This case study indicates that progress is being made, and yet, mana whenua respondents recognised that more must be done. The respondents urged further work, including that design projects should look at effective measures to be more inclusive, develop capacity and move to co-design (earlier engagement). In addition, project leaders should purposefully build opportunities into their design brief to encourage earlier engagement, training workshops and leadership beyond cultural outcomes. Invaluable opportunities remain in the evolution of urban-Indigenous design, where community stakeholders and local-Indigenous peoples can advance the workings of proactive engagement beyond cultural landscaping. Further, local government and planners have a significant role in the enhancement of Indigenous design processes by implementing and activating their obligations under te tiriti. Respondents were optimistic that the ability and capacity to engage local-Indigenous peoples in the redesign of their community streets is expanding.

Our findings add weight to previous research arguing that design processes must incorporate principles of stewardship, cultural identity, collective rights, and the political right to land and governance for Indigenous peoples in their home places [18,31,37]. As Matunga [22] and others [10,42] argue, the restitution of Indigenous materiality and memory in cities is underpinned by the recognition of Indigenous rights, authority, participation and the visible features of re-indigenising their home places in settler-colonial spaces. The respondents all spoke of these factors, reiterating that local-Indigenous autonomy is reliant on fundamental issues such as: Indigenous obligations, participation and decision making, local knowledge and learning [43]. Hence, we contend that local-Indigenous engagement and co-design is an essential strategy for improved community street and cultural landscape outcomes. In colonial cities, future urban design projects must move beyond beautification measures and instead support the continued evolution of empowered local-Indigenous autonomy. The dearth of research examining such attempts to support autonomy points to the need for further evaluation research to allow for a fuller understanding of diverse Indigenous experiences and enable more generalisable recommendations to be made. 

## 5. Conclusions

Mana whenua engagement experiences reveal the importance of developing capacity, and providing authentic and unique outcomes. While urban design has a long history of oppression and marginalisation of Indigenous peoples, projects such as Te Ara Mua demonstrate promising progress in the reclamation of cultural voice and visibility. Yet, there remains significant work to be done. Local-Indigenous autonomy is an important factor in achieving greater sovereignty and self-determination. This project reveals important steps in the evolution of urban redesign and future planning. Meaningful collaboration provides significant opportunities for the development of Indigenous capacity and the realisation of equitable co-design. This continued advancement can contribute to reconciliation and the healing of historical trauma associated with colonisation. The realisation of autonomy in community redesign projects has broader implications for Indigenous sovereignty, spatial justice and health equity. Exercising local-Indigenous autonomy offers vital potentialities to disrupt and unsettle colonising power relations as Indigenous peoples. Redesigning communities encapsulates broader potential beyond simplistic notions of making public spaces more liveable or healthier, but in fact more Indigenous and equitable. Together, urban-Indigenous design and re-indigenisation are important strategies to improve autonomy and wellbeing. Reclamation efforts led by Indigenous peoples revitalises urban spaces as Indigenous places. The future of urban design in settler cities must include not only redesigning or reimagining neighbourhoods, but also their re-indigenisation.

## Figures and Tables

**Figure 1 ijerph-18-00865-f001:**
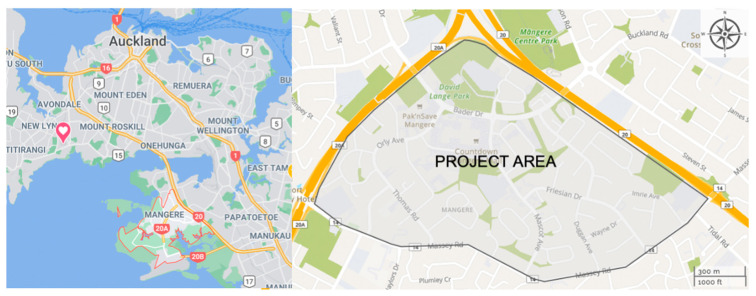
Map of Māngere—Te Ara Mua project intervention area for mana whenua engagement (Source: Google Maps & TAMFS Project Image).

**Figure 2 ijerph-18-00865-f002:**
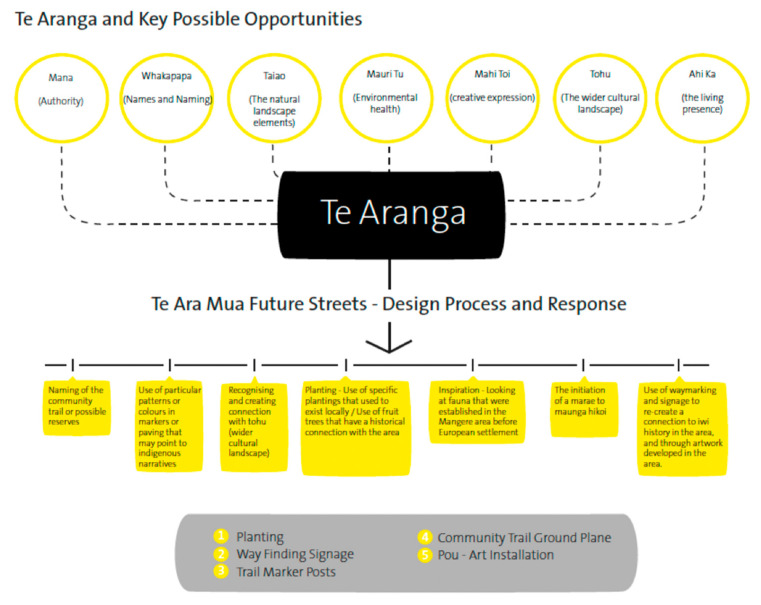
Te Ara Mua—Future Streets: Mana Whenua design plan and outputs.

**Figure 3 ijerph-18-00865-f003:**
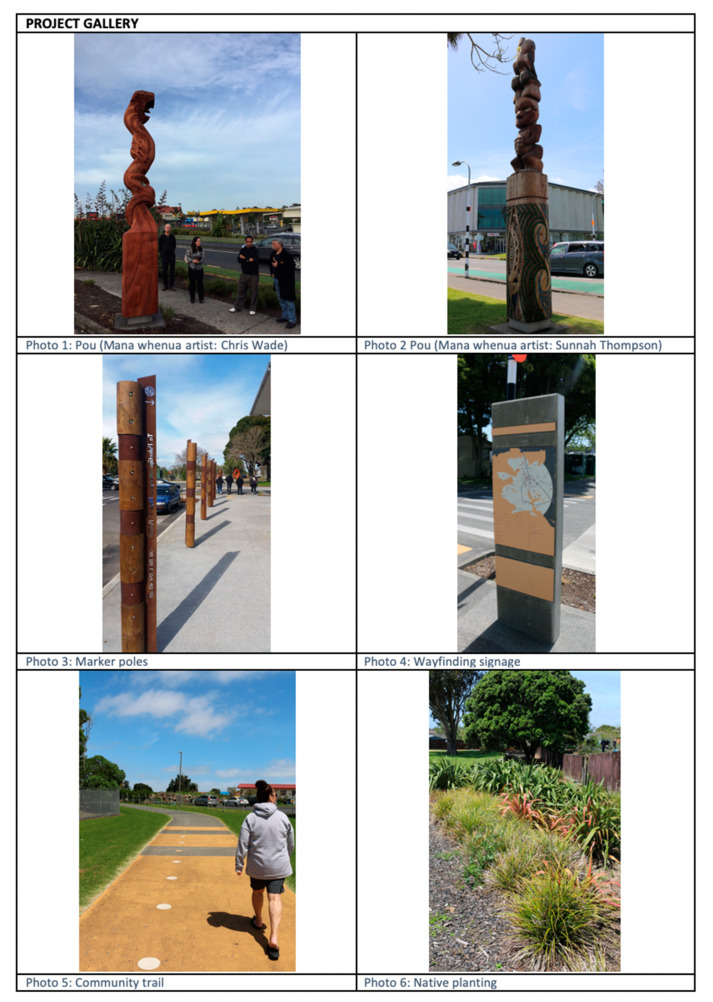
Te Ara Mua—Future Streets Photo Gallery of Mana Whenua Outputs.

**Table 1 ijerph-18-00865-t001:** Mana whenua co-design panel members with their roles in the panel and this study.

	Name	Group	Iwi, Organisation	Role	Study Contribution
1 *	Mei Hill	Mana whenua	Ngāti Whātua Ōrākei	Arts and design manager	Respondent
2 *	Berenize Peita	Mana whenua	Ngāti Te Ata	Representative	Respondent
3	XXX	Mana whenua	Te Ākitai Waiohua	Representative	Unavailable
4 *	David Thomas	Stakeholder	Auckland Council	Arts and culture project manager	Respondent
5 *	Yoko Tanaka	Stakeholder	Boffa Miskell	Landscape architect	Respondent
6	Rau Hoskins	Research team (Māori)	DesignTribe architects	Project lead, architect	Co-author
7	Adrian Field	Research team	Dovetail	Project facilitator	Co-author
8	Hamish Mackie	Research team	Mackie Research	Project lead	Advisor
9	XXX	Stakeholder	Employment change	Compliance manager	Unavailable
10	XXX	Stakeholder	Employment change	Landscape architect	Unavailable
X *	Chris Wade	Mana whenua	Te Ākitai Waiohua	Māori artist, local resident	Non-panel memberRespondent

Key: ‘XXX’ denotes members who were not able to participate in this study or consent to being named. ‘*’ identifies panel and non-panel members interviewed.

## Data Availability

Raw data will not be made publicly availability to protect the privacy of participants.

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
