# Peer review of "Local-Indigenous Autonomy and Community Streetscape Enhancement: Learnings from Māori and Te Ara Mua—Future Streets Project"

_ijerph, 2021, doi:10.3390/ijerph18030865_

Round 1

Reviewer 1 Report

This article tackles local-indigenous autonomy and community streetscape enhancement using indigenous methodology, which is a part of a wider reflection over emancipatory paradigm within qualitative inquiry, oriented at local, culturally diverse communities (McLeod, 2000).

I found this article interesting, but I have two questions and the answers should be included in the article:

  1. How many people were interviewed?
  2. Authors cite statements from only a few of them. Why? As a qualitative researcher I understand that we do not quote all responses given by the subjects, but only those that overlap with the needed information to the highest extent so it means that the selection of the sample thus expanded to the stage of presentation of research results but the authors need to mention this in Method Section.

Overall, an interesting and thoughtful paper.

Additional comments see the attachment.

Author Response

Thank you. In the table below we have itemised each recommendation and have noted action taken.

REVIEWER 1

ITEM

RECOMMENDATION

ACTION

1.

Table 1: What exactly “XXX" and "Unavailable” mean – please explain. If it means that the authors did not analyse the data from these groups, why were they in the table?

The table shows the makeup of the full mana whenua co-design panel, as well as explaining their organisational roles and the part they played in the research. Those with an XXX had either moved on or were not available for an interview as part of the study. They are denoted as XXX because they did not have the opportunity to consent to being identified by name. We’ve clarified the roles of all the panel members further in the table, and explained the XXX in the caption.

2.

Figure 3. Authors present seven photos while referring to only two of them. Why? Please explain the meaning of posting the other images.

Added explanation text of photos to new lines 220-221.

3.

Methodology: Due to the ethnographic nature of the analyses, it is worth referring to the insider/outside dichotomy.

This study was not ethnographic, but rather a standard thematic analysis of a small number of in-depth interviews. Nonetheless, we take your point about the mixed insider/outsider status of the paper co-authors, and have therefore provided some reflexive statements about this (see new lines 162-173)

4.

Line: 217: How many people were interviewed?

This has been clarified by the table and in new lines 248 and 289-290 – a total of 5 interviewees

5.

Authors cite statements from only a few respondents. Why? As a qualitative researcher, I understand that we do not quote all responses given by the subjects. Still, only those that overlap with the needed information to the greatest extent. The author needs to mention the selection of the sample, thus expanded to the stage of presentation of research results. It is also not clear to me why some of the respondents' statements are highlighted, and some are placed in the text. For example, line 302-304 and 307-310. Some of them are more important than others? Why?

This article is about prioritising and acknowledging local-indigenous autonomy. In keeping with the Kaupapa Māori methodology, we therefore purposefully used only quotes from the mana whenua participants.

Please see this explanation added in lines 292-293.

6.

Line 217: Authors said that "Individual (face-to-face) in-depth semi-structured interviews", but when I read the analyses (for example line 315) I have the impression that these were rather focused interviews.

The interviews were semi-structured, as described in the methods. We have clarified the wording that inaccurately implies focus in the interviews. We were using “concurred” as a way of saying that interviewees brought up similar points to each other rather than any implication they were concurring with the interviewer, but we can see how this could be misread.

7.

Line: 223-234 the sentence “I All respondents were offered a koha (gift, offering) for their generosity in sharing their time and knowledge for the study.” - this information is unnecessary instead of it you should indicate who was/were the interviewer(s)? What was their training? Experience? Need a sentence in about the interviewer. It is critically important in qualitative research.

Koha is a fundamental exchange or gifting for Māori of taonga (treasure including kōrero/information) – and therefore must be noted to reflect the observance of KM practices and principles in our interviews. We’ve added some Kaupapa Māori ethical principles that help to ground the way we’ve reported things in new lines 158-161.

Lines 162-173 now describes the roles, including that of the interviewer.

8.

Line 243: How did the themes emerge, strengthen, diminish, truncate, etc. What kind of themes are you talking about? You would need collation of these themes before they are discussed, which will make the article more readable. This article needs strengthening and refinement.

Lines 294-301 collate and describe the final set of five themes. We have added new lines 282-283 to explain the process of theme identification and refinement.

9.

In the interview analysis, it is always worth referring to the findings of other authors, which allows you to relate to the questions answered and discuss them.

In this Kaupapa Māori analysis, our priority was to draw meaning from local-indigenous voice both deductive and inductive, utilising a Māori framework or lens. This is in keeping with standard thematic analysis methods, in contrast to methods like critical discourse analysis, which would more usually refer to the findings of other authors as part of the results. Instead, we return to this previous work in the discussion (see lines 593-598)

The addition of Lines 158-161 also help to justify this approach.

10.

Line 285, 342, In my opinion, dots should be in square brackets […]

Our preference is not to use square brackets for cleaner, easier to read quotations. For example, respondent quote new lines 432-437 would have 5 sets of […] – however, we would be willing to defer to the journal’s own standards about this.

11.

Line 292: What these photos show, and what the respondents wanted to express through them?

We have added text explaining the use and content of photographs during the interviews (new lines 358-360)

12.

Line 330 If it's quotation marks, then where's the end of it

Amended, thanks

13.

Line 437: It also lacks a conclusion that summarises part of the analytical interviews after the last response from Mei (line 432). I suggest the authors adds two or three sentences before the discussion portion.

As per #5 - this article prioritises and honours the indigenous voice – therefore, we have deliberately given the last word of the results to an indigenous respondent.

14.

Line 473: Authors should provide the number of respondents.

Amended, thanks (line 548)

15.

I can see other limitation of this study. Authors should point that in studies based on deliberate sampling, the interpretation of results is limited to the cases studied and generalisations are not valid. It is the "internal validity" of research results (Tongco, 2007). The results of this study are not representative of any whole population, nor can they be generalised; the knowledge obtained concerns merely the analysed phenomenon in the specific context and it cannot be “transferred” (Shenton, 2004) _

While we agree that this is a micro-analysis of a single case study, within the context of a dearth of existing research on re-indigenising urban places, we nowhere claim these voices are representative of a whole population (Māori or otherwise). On the other hand, when put together with the sparse previous research, we think there remain some important lessons about process, relationships and outcomes that could be useful for other colonial settings and indigenous struggles for sovereignty, justice and wellbeing – in particular, the urgent need to move beyond tokenistic/totemic appropriation of indigenous culture in processes of “beautification”.

We’ve made some wording changes in the discussion to make it clearer where we are synthesising the findings from this case study, and where we make recommendations that put our findings together with previous work.

16.

The article contains minor inconsistencies in punctuation; some sentences made use of a serial comma, others did not. Both the en-dash and hyphen were used interchangeably as well. I have also marked sentences with larger grammatical errors in the PDF.

We have reviewed and corrected serial comma, hyphen and en-dash usage.

The sentences you highlighted have been edited for clarity. See lines 35-45

Reviewer 2 Report

"Local-indigenous autonomy and community streetscape enhancement: Learnings from Māori and Te Ara Mua-Future Street Project"

This essay should be published in its actual format.  It is well written and very clear in its goals and objectives.  It should also be remarked that the authors have taken the topic in their essay one step further by reflecting on the possible limitations they have encountered (line 471 onwards) and addressing them in a "work to be done" paragraph (line 517 and further down in the conclusion).  Moreover, the authors are fully acquainted with the difficulties involved in transitioning from well-intended and general principles about indigenous political agency —like "indigenous sovereignty, spatial justice and health equity" (line 26)—to concrete actions taken when designing urban and public spaces.  These project are inevitably subjected to many restrictions in their practical implementation.  The inclusion of these two aspects in the essay speaks highly of the authors.

I have no comments to make in relation with the sections in which this article has been divided.  However, I would like to make a few comments regarding the essay even though I am sure the authors have also considered these aspects.

I wonder if it would be helpful to engage in a theoretical discussion to better define concepts like indigenous cultural identity (line 19), indigenous sovereignty (line 26) or re-indigenising (line 28).  Despite the fact that defining these concepts does not fall within the scope of this paper, it will not be detrimental to know exactly what the authors understand by them in order to fully evaluate the Te Ara Mua-Future Street Project.  On the one hand, it seems reasonable to use these terms to stress how the participation of local indigenous communities undermines the hegemonic presence of cultural and spatial practices related to settlers and colonizers.  The authors themselves emphasizes this aspect in line 113.  It is already well-known how European and other colonizers of other ethnicities frequently denigrated and suppressed cultural expressions connected with indigenous groups.  In this sense, I would argue that the use of these terms is fully adequate. 

On the other hand, we might run the risk of essentializing indigenous belonging by excluding other community groups who might also share the same interests in health equity and spatial justice.  If, when trying to increase indigenous participation in urban designs, we take "indigenous" to be a homogenous ethnical classification, we could be doing a disservice to these groups because it ignores differences within the same ethnic group.  In this guise, every effort should be made to repeal a type of cultural identity that reduces membership to mere numbers without any other qualification.  How to revitalize a cultural identity without falling into essentialisms has to be at the center of any participatory process that organizes around ethnic lines.  This must be done regardless how important it is to emphasize ethnic differentiations in a context of centuries-old discrimination against indigenous peoples.  In this way, I wonder how the term re-indigenising is perceived by indigenous groups.  Are they in favor or against it?  

Since some of the participants interviewed fall prey to essentializing indigeneity (line 266, 277), I am wondering how indigenous belonging could erase—even if this is not its intention—inequalities related to gender or social class within the same community.  The presence of indigenous members in the decision making process is not guarantee enough to avoid falling into sexist affirmations, for example.  The result could lead to the dismissal of indigenous women's claims related to gender inequalities within their own ethnic group.  The same could be said about social class: in all probability, inside the same indigenous groups there are substantial differences in income.  In the same vein, indigenous participation in urban planning will always be very limited if no consideration is given to how non-indigenous individual property rights are detrimental to indigenous community based property.

Finally, I would like to congratulate the authors for their paper.  It has been a pleasure to read it.   

Author Response

Thank you. In the table below we have itemised each recommendation and have noted action taken or our response.

REVIEWER 2

ITEM

RECOMMENDATION

ACTION/RESPONSE

1.

I wonder if it would be helpful to engage in a theoretical discussion to better define concepts like indigenous cultural identity (line 19), indigenous sovereignty (line 26) or re-indigenising (line 28).  Despite the fact that defining these concepts does not fall within the scope of this paper, it will not be detrimental to know exactly what the authors understand by them in order to fully evaluate the Te Ara Mua-Future Street Project.  On the one hand, it seems reasonable to use these terms to stress how the participation of local indigenous communities undermines the hegemonic presence of cultural and spatial practices related to settlers and colonizers.  The authors themselves emphasizes this aspect in line 113.  It is already well-known how European and other colonizers of other ethnicities frequently denigrated and suppressed cultural expressions connected with indigenous groups.  In this sense, I would argue that the use of these terms is fully adequate. On the other hand, we might run the risk of essentializing indigenous belonging by excluding other community groups who might also share the same interests in health equity and spatial justice.  If, when trying to increase indigenous participation in urban designs, we take "indigenous" to be a homogenous ethnical classification, we could be doing a disservice to these groups because it ignores differences within the same ethnic group.  In this guise, every effort should be made to repeal a type of cultural identity that reduces membership to mere numbers without any other qualification.  How to revitalize a cultural identity without falling into essentialisms has to be at the centre of any participatory process that organizes around ethnic lines.  This must be done regardless how important it is to emphasize ethnic differentiations in a context of centuries-old discrimination against indigenous peoples.  In this way, I wonder how the term re-indigenising is perceived by indigenous groups.  Are they in favour or against it?

How to revitalize a cultural identity without falling into essentialisms has to be at the centre of any participatory process that organizes around ethnic lines. 

Thanks very much for these thoughtful comments, presumably for our consideration only, since your recommendation is kindly to publish in the current format.

It was for the reasons argued here that we took the space to explain what we mean by these terms in lines 55-98. We consider that any further theoretical explication and discussion would unbalance the paper and detract from the central scope.

In response to the final question here: “In this way, I wonder how the term re-indigenising is perceived by indigenous groups.  Are they in favour or against it?”: this questions contradicts your musings on homogenisation in the next box, by being homogenising in itself! The indigenous authors and participants involved in this paper obviously argue for re-indigenising as a helpful concept, alongside other indigenous authors quoted.

In response to the comments about “essentializing indigenous belonging”: we are not talking about belonging in a “community” sense in this paper, rather, as explained, we are viewing “re-indigenising” as responding to historical alienation from land within a bicultural Treaty context (which we make clear in our Introduction), wherein iwi never ceded sovereignty over that land.

In a general sense, therefore, the results of upholding the Articles of this Treaty on “other community groups who might also share the same interests in health equity and spatial justice” is a secondary issue that is of little relevance to this paper. Though as a complete out of scope aside, it’s our experience that there is growing understanding of the likely co-benefits for these communities, including via re-enabling the expression of other widely held Māori values, such as manaakitanga (the expression of generous hospitality).

2.

If, when trying to increase indigenous participation in urban designs, we take "indigenous" to be a homogenous ethnical classification, we could be doing a disservice to these groups because it ignores differences within the same ethnic group.  In this guise, every effort should be made to repeal a type of cultural identity that reduces membership to mere numbers without any other qualification. 

We agree with these comments about avoiding homogeneity assumptions. We have tweaked some wording in the discussion to ensure that we’re not implying a unified indigenous voice. In addition, we think the naming of participants and description of their roles avoids the “type of cultural identity that reduces membership to mere numbers without any other qualification” you’re referring to here.

Reviewer 3 Report

The paper addresses and explores the complex issue of the link between urban design and healthier communities, providing tangible evidence from the case study. 

The Te Ara Mua-Future Street Project as well as demonstrating that co-design has great potential in restoring autonomy and revitalizing indigenous cultural identity, but also traces a way to finally make effective projects of intercultural integration in urban areas. The study also shows that the performance of such projects is a determining factor for the spatial justice and health equity of the entire urban space and the total community that dwells in it. The work, in addition to being clear and well structured, makes an important contribution to enhancing the voice and visibility of indigenous people, reaffirming their presence and ownership of traditional lands despite the processes of urbanization and colonisation. 

Despite the limitations highlighted by the authors themselves (the one-time interview and the size of the sample analyzed), the work makes a concrete contribution to the design of urban spaces and to the development of urban regeneration hypotheses based on a proactive engagement of indigenous communities. I am therefore in favor of the publication, not considering it necessary to revise the English language and appreciating the readability of the work as a whole.

I recommend supplementing the paragraph of the conclusions with a reference to future developments in research, both in terms of overcoming the limitations that emerged in this first study and in terms of evaluating and monitoring the success of the proposed measures.

Minor suggestions concern:
- in figure 1 the representation scale and the north direction arrow are missing. I would also add a framing map to understand which area has been zoomed on

Author Response

Thank you. In the table below we have noted the reviewers recommendations and our actions or response.

REVIEWER 3

ITEM

RECOMMENDATION

ACTION

1.

I recommend supplementing the paragraph of the conclusions with a reference to future developments in research, both in terms of overcoming the limitations that emerged in this first study and in terms of evaluating and monitoring the success of the proposed measures.

We have added a sentence to the end of the Discussion (new lines 604-606) about further research.

2.

Minor suggestions concern: in figure 1 the representation scale and the north direction arrow are missing. I would also add a framing map to understand which area has been zoomed on

Direction arrow and scale now added; as well as a map of Auckland showing where the intervention area lies within the region.